METHODS AND RESOURCES

# Blik is an extensible 3D visualisation tool for the annotation and analysis of cryo-electron tomography data

**Lorenzo Gaifas**[1]*, **Moritz A. Kirchner**[1], **Joanna Timmins**[1], **Irina Gutsche**[1,2]*

**1** Institut de Biologie Structurale, Université Grenoble Alpes, CEA, CNRS, IBS, Grenoble, France,
**2** Department of Chemistry, Umeå University, Umeå, Sweden

* lorenzo.gaifas@gmail.com (LG); irina.gutsche@ibs.fr (IG)

**Data Availability Statement:** The current version of blik at time of publication is available permanently at Zenodo (https://doi.org/10.5281/

## Abstract

Powerful, workflow-agnostic and interactive visualisation is essential for the ad hoc, human-in-the-loop workflows typical of cryo-electron tomography (cryo-ET). While several tools exist for visualisation and annotation of cryo-ET data, they are often integrated as part of monolithic processing pipelines, or focused on a specific task and offering limited reusability and extensibility. With each software suite presenting its own pros and cons and tools tailored to address specific challenges, seamless integration between available pipelines is often a difficult task. As part of the effort to enable such flexibility and move the software ecosystem towards a more collaborative and modular approach, we developed `blik`, an open-source `napari` plugin for visualisation and annotation of cryo-ET data (source code: https://github.com/brisvag/blik). `blik` offers fast, interactive, and user-friendly 3D visualisation thanks to `napari`, and is built with extensibility and modularity at the core. Data is handled and exposed through well-established scientific Python libraries such as `numpy` arrays and `pandas` dataframes. Reusable components (such as data structures, file read/write, and annotation tools) are developed as independent Python libraries to encourage reuse and community contribution. By easily integrating with established image analysis tools—even outside of the cryo-ET world—`blik` provides a versatile platform for interacting with cryo-ET data. On top of core visualisation features—interactive and simultaneous visualisation of tomograms, particle picks, and segmentations—`blik` provides an interface for interactive tools such as manual, surface-based and filament-based particle picking, and image segmentation, as well as simple filtering tools. Additional self-contained napari `plugins` developed as part of this work also implement interactive plotting and selection based on particle features, and label interpolation for easier segmentation. Finally, we highlight the differences with existing software and showcase `blik`'s applicability in biological research.

## Introduction

Cryo-electron tomography (cryo-ET) is a powerful three-dimensional (3D) cryo-electron microscopy (cryo-EM) imaging technique for visualisation and structural analysis of biological

zenodo.10894490). The source code is maintained at https://github.com/brisvag/blik.

**Funding:** This project received funding from GRAL, the Grenoble Alliance for Integrated Structural and Cell Biology, a programme of the Chemistry Biology Health Graduate School of Université Grenoble Alpes (ANR-17-EURE-0003), and from the Agence Nationale de la Recherche (grant ANR DecRisp ANR-19-CE11-0017-01 to I.G.). The funders had no role in study design, data collection and analysis, decision to publish, or preparation of the manuscript.

**Competing interests:** The authors have declared that no competing interests exist.

**Abbreviations:** 3D, three-dimensional; cryo-EM, cryo-electron microscopy; cryo-ET, cryo-electron tomography; GUI, graphical user interface; IO, input/output; ML, machine-learning; SDF, signed distance field.

samples in situ [1]. In recent years, rapid development of software for cryo-ET data processing and analysis has brought great advances in tools for tilt series alignment [2], particle picking [3,4], averaging and classification routines [5–7], denoising [8,9], and more [10]. Thanks to subtomogram averaging, cryo-ET is now routinely used to determine the structure of biological macromolecules in situ, achieving in the most favorable cases sub-nanometer resolutions [11,12].

While more and more powerful, existing workflows still rely on extensive human intervention due to the high heterogeneity of requirements and samples [13,14]. Powerful and user-friendly visualisation tools are needed for effective human-in-the-loop pipelines to minimise the friction at the human–machine interface, and should be composed of modular and extensible software, to maximise reusability and simplify integration of different existing toolkits.

A common practical challenge encountered by scientists working on cryo-ET data is indeed the (in)compatibility between different software tools. Some of the most widespread cryo-ET software suites (such as IMOD [15], PEET [16,17], Relion [18], Dynamo [19], and emClarity [20], and PyTom [21,22]) all use different file formats for particle poses and tilt series metadata. This constitutes a barrier for users who need to use features from different tools on the same data: at best, users might miss out on important features from other software; at worst, integration may silently go wrong and cause issues in later steps. Entire software suites such as Scipion [23] are devoted to integrating normally incompatible cryo-EM and cryo-ET software into pipelines.

With useful features scattered among different programs (e.g., AreTomo's unsupervised alignment [2], Topaz's denoising [8], CrYOLO's filament picking [4], EMAN2's trainable segmentation [10]), and numerous small custom scripts developed by researchers tailored to a specific project's needs (distributing or selecting particles, improving alignments, etc.), software integration is a real and common concern.

Even when a compatible tool is available or conversion possible, it is often hard to tell when it worked properly due to lack of generalised visualisation tools for inspecting and validating data throughout the pipeline. Due to the aforementioned compatibility restrictions, popular software suites (such as IMOD and Dynamo) often provide built-in visualisation tools, duplicating development efforts and further deepening the separation between pipelines.

Finally, many existing visualisation tools are not easily hackable by users to extend them with custom functionality. Even those that offer ways to extend their functionality (such as ChimeraX [24] through its Python API or Dynamo with MATLAB [25] code), provide limited interface to data and rendering code, or require considerable programming skills to do so.

To address these issues, we present blik, a new software for interactive visualisation, manipulation, and analysis of cryo-ET data. The code is open-source and welcomes community contributions at https://github.com/brisvag/blik. blik is a plugin for napari [26,27] (https://napari.org/), a visualisation software focused on scientific imaging, with data segmentation and annotation available as core features. It has both a programmatic and a graphical interface, allowing for seamless integration of interactive visualisation and scripted pipelines. napari offers great customisation options, powerful built-in tools, and a growing plugin ecosystem, which allows blik to focus on specific cryo-ET needs.

To address the challenges listed above, blik's design choices, features, and architecture reflect the following primary goals:

- **Compatibility:** blik can read and write data in file formats from a variety of different software suites, including IMOD, Relion, and Dynamo. This makes it easy to switch between tools or to integrate custom scripts into a workflow.

- **Interactivity:** blik provides interactive visualisation that allows users to explore data programmatically and visually at the same time. Data is always accessible through a standard

Python console and in simple, well-established formats such as `numpy` arrays and `pandas` dataframe. This makes it easy to validate data during processing and to identify problems as soon as they arise, as well as to provide a framework for quicker prototyping and debugging of new workflows.

- **Hackability:** `blik` is open-source and easy to extend with custom functionality. This allows users to tailor the software to their specific needs. The `napari` plugin ecosystem also allows taking advantage of many existing analysis tools, even from different imaging fields. Additionally, `blik`'s input/output (IO) capabilities are easily extensible by users to include new custom formats.

- **User-friendliness**: blik's, and most of napari's, functionality are also exposed in the graphical user interface (GUI), making it also easy to use for non-programmers.

- **Performance:** Thanks to `napari`'s visualisation backend `vispy`, `blik` has performant rendering which can handle large 3D (and more) datasets, even larger-than-memory thanks to `dask`.

- **Community**: To foster community contribution, code reuse, and jumpstart other projects, many contributions were upstreamed to `napari`, `vispy`, or extracted into simple single-use libraries usable by other projects (`teamtomo`).

The use of Python for the development of `blik` and `napari` is crucial to further these goals. In the last years, the integration of scientific Python in high school and university educational curricula played a pivotal role in its democratisation. The surge of popularity of the scientific Python ecosystem with modular and reusable tools is largely due to its versatility, readability, simplicity, extensive documentation, and a wide community support.

This makes `blik` a convenient entry point for cryo-ET-interested newcomers and lowers the barrier for the creative leveraging of basic programming skills for their everyday research and applications, enabling individuals with varying levels of programming experience to quickly grasp and implement solutions.

Moreover, the scientific Python ecosystem is well-established in many imaging fields, and is becoming a player in the cryo-EM and cryo-ET world. By adhering to the practices and conventions of this ecosystem, `blik` allows to easily integrate many available field-agnostic tools (e.g., `scikit-image` and `scipy` for image and annotation processing, `pytorch` and the plethora of Python machine-learning (ML) tools for several types of analysis), take advantage of existing solutions and avoid the tendency to "reinvent the wheel" that scientific software can often be prone to.

Finally, to help users to seamlessly integrate `blik` into their existing workflow, `blik` is currently being integrated into Scipion as a plugin (https://github.com/scipion-em/scipion-em-blik).

## Results

The following sections describe all the features provided by `blik` and their implementation details. To help users get started with `blik`, we provide a supplementary tutorial that explains the `blik` user interface and offers practical guidance on using each tool in a cryo-ET workflow (S1 Tutorial). This tutorial, together with a more comprehensive and up-to-date documentation, is hosted at https://brisvag.github.io/blik/.

### Visualisation

`blik` relies on `napari` for performant visualisation. The `napari` core is field-agnostic and requires the development of custom readers and writers to convert specific file formats into a

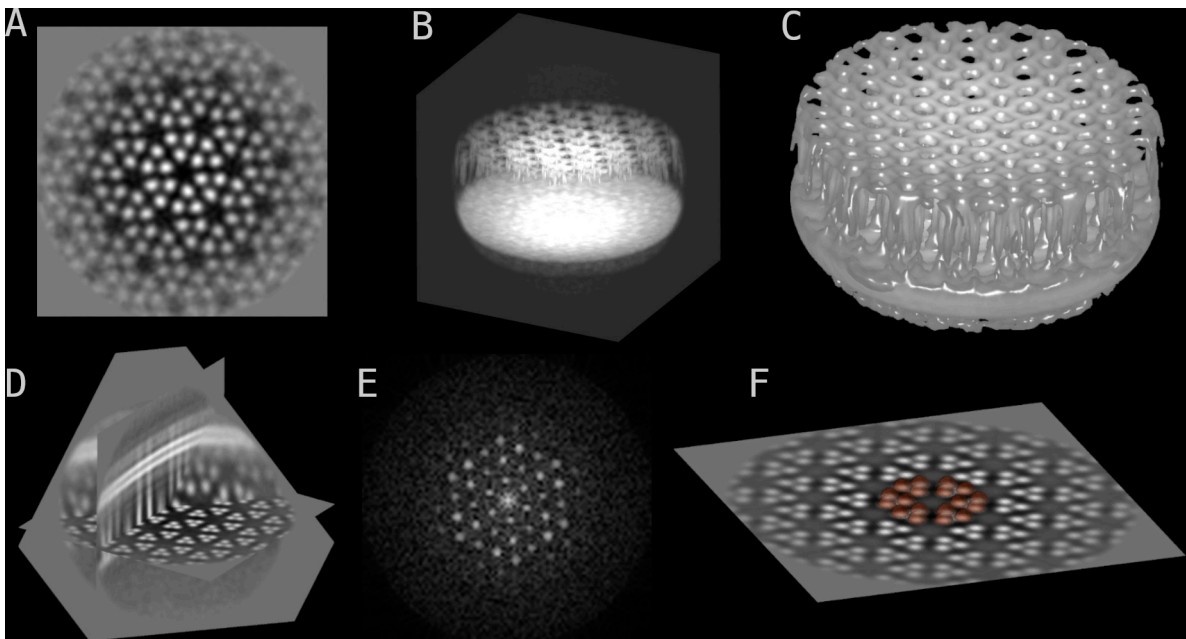

**Fig 1. Volume visualisations.** Showcase of the image visualisation capabilities of `napari` and `blik`. Chemoreceptor arrays of the *E. coli* minicells [28] are used for illustrative purposes throughout the figures. **(A)** 2D slice through a 3D volume. **(B)** Maximum intensity projection of the 3D volume. **(C)** Isosurface. **(D)** Arbitrary 3D plane slicing in 3D view. **(E)** 2D slice through the 3D power spectrum of the volume. **(F)** Segmentation (in semi-transparent brown) of 3D objects.

`napari` visualisation. This has been an important focus of `blik` so far, through the development of `cryohub` and the napari representation of particle poses as oriented points.

Data in napari is exposed as layers that can be controlled individually (similarly to general image processing software like GIMP or Adobe Photoshop). There are several types of layers; the main ones used by blik for visualisation purposes are Image for images and volumes, and Points and Vectors for particle poses.

**Images and volumes.**   Images, image stacks, and volumes in the most common formats (`.tif`, `.mrc`, `.em`, `.hdf`) can be opened and viewed both in 2D as slices (Fig 1A) and in 3D as volumetric projections (Fig 1B), isosurfaces (Fig 1C), and more. It is possible to change colormaps, contrast limits, gamma, and other basic image visualisation parameters. 3D visualisation also allows for slicing the volumes at arbitrary planes (Fig 1D), similar to `IMOD`'s `slicer` tool.

`blik` also provides some widgets with extra functionality to complement image visualisation:

- A basic GPU-accelerated filtering tool (gaussian blur) for 2D images and 2D slices that is computed on the fly and whose parameters can be regulated with sliders. Simple gaussian filters are frequently used in image visualisation, especially with noisy cryo-EM data. While generating a filtered image is a relatively fast procedure, it is rarely fast enough to be computed on the fly on a CPU. GPU-accelerated filtering allows switching on and off on the fly, as well as changing kernel size and sigma, without having to generate a new image. While `blik` exposes only gaussian filtering, the underlying logic was implemented in the `vispy` OpenGL shader allowing for arbitrary convolutional kernels. Firstly, the image texture is sampled in an NxM grid (for a kernel of shape NxM) centered around the texture

coordinates. A weighted average is then computed based on the kernel weights and the resulting value is forwarded to the rest of the shader.

- A `power spectrum` widget, which quickly computes the power spectra of single images, stacks, or volumes (Fig 1E). Power spectra are a fundamental tool for data inspection and validation in cryo-EM, from estimating resolution, to generating hypotheses about symmetry, to finding caveats in the data collection procedure. Like any other computation made by `blik` or `napari`, the power spectrum is then available as a normal `numpy` array for further use from within Python or to export to the available formats.

Additionally, masks and pixel-based segmentations (often called just "segmentations" or "labels" in `napari` jargon) are easily displayed (and modified with all the tools provided by `napari`, such as free-hand painting, even in 3D) by using the `napari` Labels layer, which is designed specifically to work with segmentation data (binary or integer arrays) (Fig 1F). Label processing is a particularly thriving area of the `napari` ecosystem, albeit mostly in the field of fluorescence and optical microscopy. This is a great opportunity for knowledge sharing and code reuse between fields that otherwise are typically relegated to separate software pipelines.

**Particles.** Particle data (coordinates, orientations, and any additional features and metadata) can be loaded from common formats (`.star`, `.tbl`). Coordinates are visualised as spheres, and orientations as basis vectors centred on the spheres (Fig 2). Both components can be disabled or tweaked (such as color-coded or resized, labeled, etc.) for better visualisation. As with images, particles can be easily viewed in 3D.

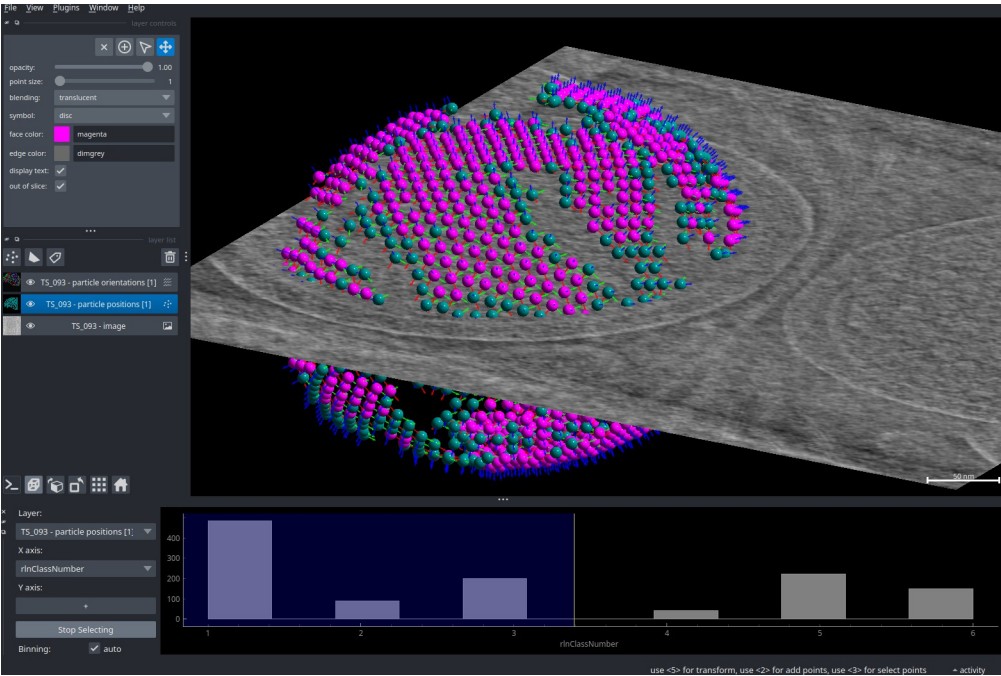

**Fig 2. 3D particle visualisation and feature plotting.** Particles are from a `Relion` 3D classification of chemoreceptor arrays of the *E. coli* minicells [28]. Orientations are displayed with the 3 basis vectors in red/blue/green. The properties-plotter widget (bottom) shows a plot of a `Relion` data column—in this case, the `rlnClassNumber` —as a histogram. By selecting an area of this plot, the corresponding subset of particles is selected. This can be then used for further processing; in this case, selected particles are coloured differently (in magenta), showing that the selected classes contain mostly particles from the core of the chemoreceptor array lattice.

Since particles are actually simple `napari` Points layers, everything in the `napari` ecosystem that works with points will work with particles, such as manual or automated selection based on features, editing and coloring, classification, etc. Once again, the wealth of cross-field contributions towards the `napari` ecosystem is a valuable asset for processing data.

For example, points (and thus particles) may hold extra metadata in their `features` dataframe, such as classification results or quantitative values such as a confidence score. All such features can be used to encode visualisation parameters like color, rendered symbol (sphere, square, cross, etc.) and size, facilitating data inspection and selection of suitable candidates for further processing.

To fully take advantage of this, shipped as part of `blik`—but developed as a standalone field-agnostic `napari` plugin—is the `napari-properties-plotter` plugin, which allows interactive plotting of per-point features (Fig 2, bottom widget), such as Relion's figure of merit, classification results, resolution estimates, etc. Distribution histograms or scatter plots with any combination of features as axes can be automatically generated by simply selecting the desired features. Moreover, by picking subsections of such plots, particles can then be selectively rendered, modified, saved, and otherwise processed.

## Input/Output

The reading, writing, and conversion logic used by `blik` was developed as a standalone library, `cryohub`. This library has a modular design to allow reuse in other applications and simplify the contribution of new formats. Currently, it supports the following formats for images and segmentations:

- `.mrc` (and the `.mrcs`, `.st`, `.map`, `.rec` variants)
- `.tif(f)`
- `Dynamo.em`
- `EMAN2.hdf`

and the following formats for particles:

- `Relion.star` (Relion $\geq$ 3.0)
- `Dynamo.tbl`
- `CrYOLO.cbox` and `.box`
- `EMAN2.json`

Where possible, `blik` makes use of lazy loading via `dask` [29], which allows working with data that is larger-than-memory and loading full datasets at once for ease of browsing.

## Image segmentation

A few annotation and analysis tools were implemented as part of `blik` or standalone `napari` plugins.

The `napari` community has already developed numerous plugins for segmentation and annotation of 2D and 3D imaging data, spanning from manual annotation and traditional image-processing-based segmentation to AI tools. `blik`'s current contributions to this ecosystem are in some cases general-purpose—such as utilities for manual annotation of volumetric segmentations—and in other cases focused on cryo-ET-specific issues that are not easily solved

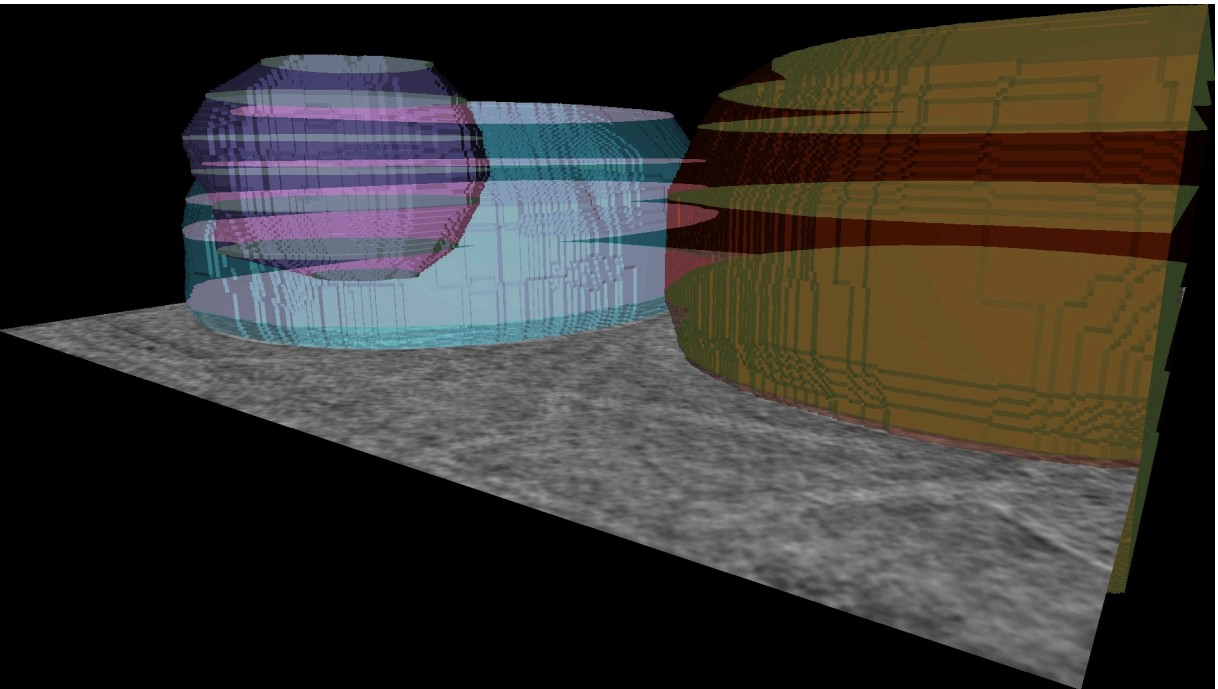

**Fig 3. Interpolated segmentations.** A tomogram is manually annotated on a few 2D slices using the `napari` labels painting functionality. These sparsely annotated 2D slices are then interpolated with the label interpolator widget to generate a volumetric segmentation of multiple objects.

by existing methods—such as rigorous geometry-based particle picking on filaments and surfaces.

Pixel-based image segmentation is already natively supported in `napari` through the `Labels` layer, with mature tools for editing, data exploration, and annotation. One previously missing feature is the ability to easily interpolate labels from sparsely annotated 2D slices of a 3D volume into full 3D volumetric labels. `blik` brings a standalone plugin for this purpose called `napari-label-interpolator` (Fig 3), which works with any `Image` layer and interpolates n-dimensionally (e.g., it can interpolate 3D labels over a time series).

The plugin works by interpolating signed distance fields (SDFs) across multiple n-dimensional slices. For each annotated slice along the interpolation dimension and for each label, the (n-1)-dimensional SDF is computed. SDF slices are then linearly interpolated with a simple weighted average, with weights proportional to the distance from the neighboring annotated slices. Where the weighted average is greater than zero, the voxel is considered to be part of the interpolated label.

This system plays very well with volumetric annotation, but has limits with thin labels such as filaments or surfaces. We plan to add different types of interpolation in order to widen the scope of application and usability of this tool.

### Filament and surface generation

A different kind of image segmentation uses mathematical representations such as filaments, surfaces, and fields to describe morphological features. In this context, `blik` contributions were made to a general-purpose library at morphometrics/morphosamplers, which aims to provide generalised morphological descriptions and sampling for scientific imaging. Specifically, we developed a helical filament model based on parametric splines, as well as a

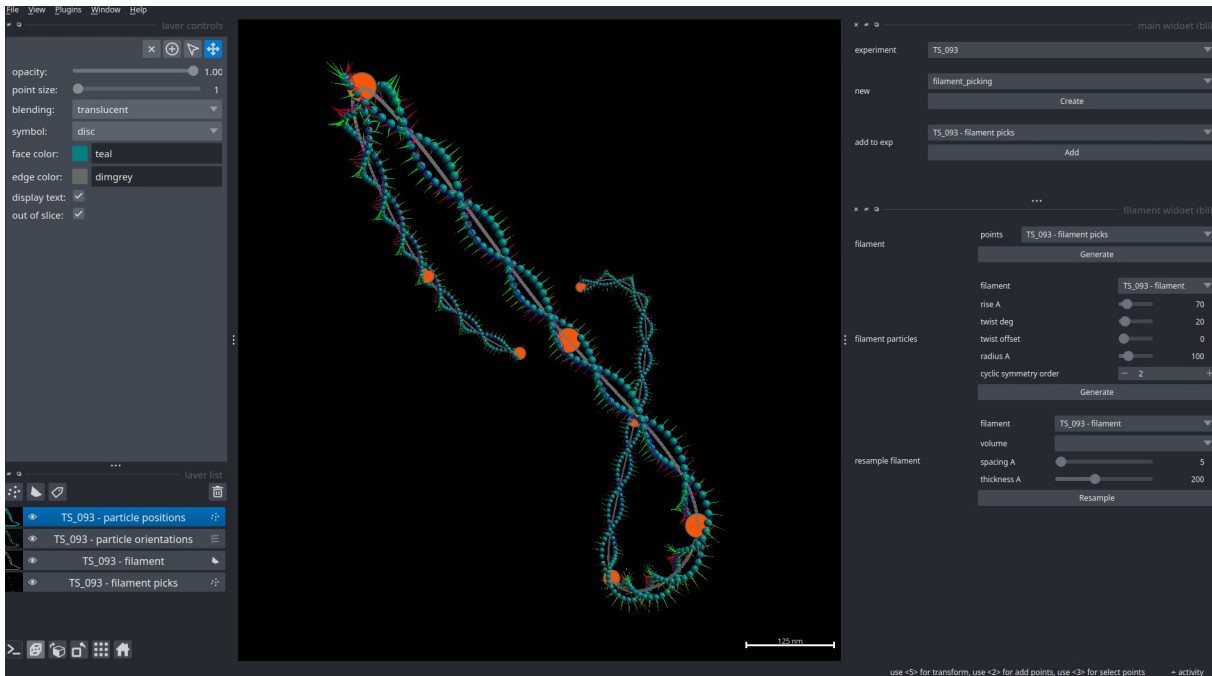

**Fig 4. Filament-based particle picking.** Particle picks along a helical filament generated by `blik`. Starting from simple points picked manually along the desired path (orange points) and using the filament widget (on the right), a spline representation is computed (grey) and used to generate particle poses (cyan spheres with basis vectors) according to the given helical parameters defined in the widget.

parametrised spline-grid surface model which allows to rigorously annotate surface-like objects such as membranes from simple point annotations.

The `morphosamplers` code was implemented with 2 main goals: keeping the manual annotation procedure simple and robust, and generating ordered, regularly spaced poses that can be used both for particle picking for subtomogram averaging and to allow volume resampling (see Resampling). Picking particles as a regularly spaced lattice can significantly improve the results of subtomogram averaging when it reflects the underlying geometry of the sample [13,30]. Model picking tools in `Dynamo` [19] had a strong influence on the purpose and functionality of `blik`; unlike the `Dynamo` mesh-based approach, the spline-based implementation aims to optimise regular Euclidean spacing and consistency in initial particle positioning and orientations.

**Filaments.**   Filament picking (Fig 4) uses a relatively simple single-spline approach:

- Points are picked manually in 3D space along the filament.

- A spline representation is generated, parametrised so that samples are equidistant in Euclidean space.

- Given a specific rise and twist, particles are generated along the spline in a helical pattern.

- If a radius is given, the particles are shifted away from the spline by that amount.

- If a symmetry group is given, symmetric copies are created around the filament axis.

**Surfaces.**   The surface generation uses the same underlying parametric spline logic as filaments, but creates a grid of splines to capture the surface shape (Fig 5). The procedure can be broken down into the following steps:

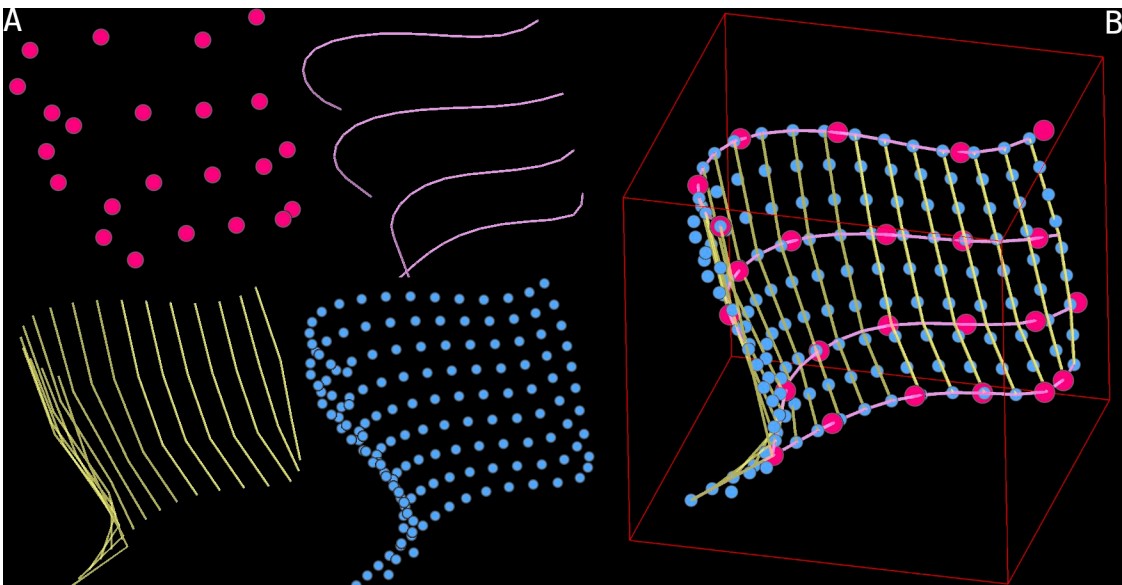

**Fig 5. Surface generation procedure. (A)** Step-by-step procedure used by `blik` to generate surfaces. An initial set of points (magenta) is picked on a few Z-slices following the desired surface. A first set of splines (purple) is generated within each Z-slice, and then used to generate equidistant samples which become the nodes of the next set of splines, perpendicular to the first (yellow). The second set of splines is then resampled with the same spacing, thus generating a final set of grid-like, evenly spaced samples. **(B)** All the steps combined.

- A few points are picked along the desired surface, repeating at different Z-slices.

- For each Z-slice annotated as such, a parametric spline (with desired interpolation order and smoothing) is computed.

- Points are distributed uniformly on these splines, ensuring equidistance in Euclidean (and not parametric) space.

- The resulting strips of points are then aligned by minimising index-wise distances and padded to enforce a rectangular grid.

- New splines are then computed by using the generated points as control points. This results in parallel splines, perpendicular to the first set and equidistant from each other.

- New points are then generated on this second group of splines, similarly to the first round, equidistant in Euclidean space. This results in a full rectangular grid of equidistant points.

- A third group of splines is generated in the same orientation as the first group.

- The second and third group of splines are used to compute the derivatives; this gives—for each point on the surface—2 vectors tangent to the surface and perpendicular to each other. With these, normal vectors are calculated, giving the full orientation of each particle on the surface.

 Parameters such as interpolation order, inter-particle distance, and smoothing can be controlled, allowing fine-tuning of the generated surface.

 While this approach is advantageous for relatively regular surfaces, for which the generated grid of splines provides uniformly spaced and oriented points on the surface, its main

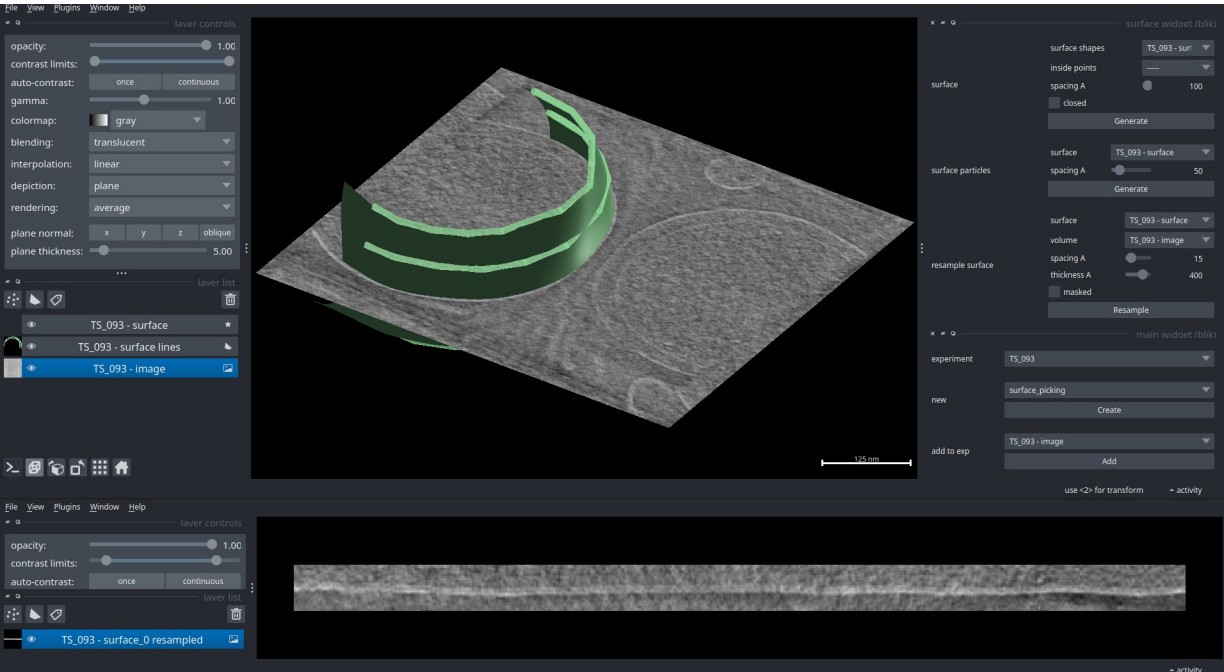

**Fig 6. Surface-based volume resampling. Top**: A surface is generated using the surface widget, following the shape of a minicell membrane. The tomogram volume is then resampled based on the picked surface and a few parameters set in the widget, such as sample thickness and pixel spacing. **Bottom**: The result is a 3D volume containing a "straightened" version of the picked surface, displayed here averaged along Z into a 2D image.

downside is that complex surfaces will inevitably deform the grid. This method struggles especially with strongly irregular surfaces, such as membranes with numerous invaginations located at different Z-slices.

**Resampling.** The grid-like, regularly spaced nature of these models can be additionally used to create visualisation aids (meshes and filaments) and to resample the annotated volume along the annotated object (Fig 6). Such volume resampling can be useful for quantitative and spatially consistent volume analysis of otherwise complex 3D objects; density profiles of a complex 3D surface can thus be generated while retaining spatial information. In practice, this can also be used to aid visualisation, by "straightening" an otherwise curved surface.

## Particle picking

`blik` provides a few tools for picking particles for subtomogram averaging. All such particles can then be saved in the formats implemented by `cryohub` for further processing.

The most basic picking tool is a manual picker; simply clicking points on an image or volume slice will generate a particle in that position. Manually modifying the orientation is not yet implemented, but is planned for a future release.

For more complex picking, the aforementioned filament and surface models can also be used to generate particles (Fig 7). These will be regularly distributed on the surface with a provided inter-particle spacing and oriented with their Z basis vector along the filament axis, or the normal of the surface. This is useful for initialising particle picks for objects following an underlying geometry, such as helical filaments and membrane proteins, and is particularly suited for dense lattices.

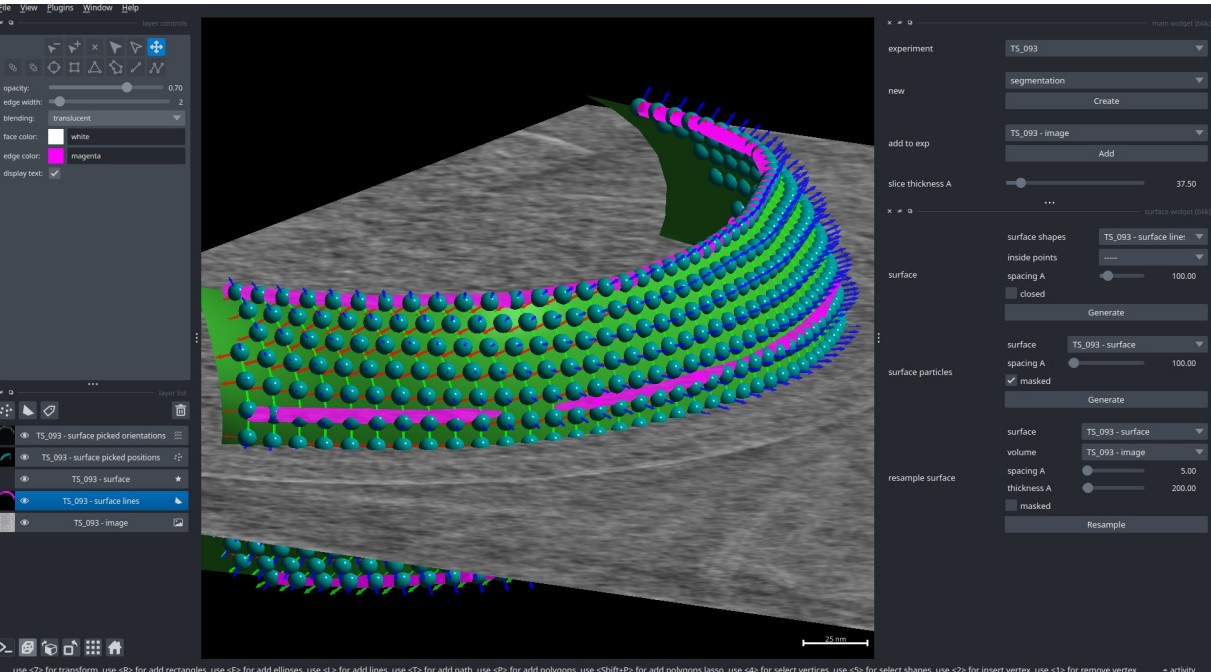

**Fig 7. Surface mesh and particle picks.** A surface is generated using the surface widget, starting from the manually picked lines (magenta) which follow the shape of a chemosensory array. The generated surface is displayed as a mesh (green). Particles for subtomogram averaging (cyan spheres and red/green/blue basis vectors) are then generated based on a few parameters set in the surface widget (on the right).

## Scipion integration

To ease the integration of `blik` into existing workflows, we also release a `blik Scipion plugin`. This plugin exposes all of `blik`'s main functionalities (visualisation, segmentation, and particle picking) as Scipion protocols that can be used in combination with all the existing tomography protocols provided by Scipion and other plugins.

## Discussion

With `blik`, we add to the cryo-ET software ecosystem an integratable and interactive data viewer, radicated in the Python ecosystem, and a reusable and extensible set of libraries and tools for annotation and picking. There are several existing tools with similar goals and functionality to `blik`; in this section, we discuss the most widely used options known to us, examine their compatibility with `blik`, and compare their features with our tool. This should provide a starting point for the reader to choose the appropriate tool for their project.

`blik` was tested primarily to work within the `Relion&Warp` pipeline [7,13,18], but was designed to be workflow-independent and thanks to cryohub easily extensible with new data formats. Following this workflow, other than the ubiquitous `.mrc` image format and `Relion's .star` particle format, one of the first compatibilities to be developed was with image and particle data from the `MATLAB` software suite `Dynamo` [19] and its processing pipeline. Dynamo provides its own visualisation and picking tools, which were also the main inspiration for the geometry-based filament and surface particle generation in `blik`. While particle picking in blik has currently a smaller scope compared to `Dynamo`, its implementation in Python is intended to be more easily maintained and extended in the future by users, while leveraging the full-featured `napari` visualisation.

`IMOD` [15] is arguably the giant in the field, with a long history of development and a wealth of features for image processing and annotation. Given its extensive capabilities and the constant development, it is usually the first choice when it comes to cryo-EM/ET processing, visualisation, and annotation. Many of these features are very useful, and we plan to add support for them in `blik` or `napari` in the future. Integration between `blik` and `IMOD` is seamless for most common operations, such as working with data processed through `IMOD`'s tomography pipeline etomo. `IMOD`'s 3d viewer `3dmod` is full-featured and fast, and has several modes for 2D projection and slicing. It has however a more limited 3D renderer compared to `napari` (no volumetric projections, slicing planes in 3D view, or native particle pick viewer), which limits its applicability for the inherently 3D work of tomography. Being written in C, it is also nontrivial to extend for users with limited programming knowledge.

Thermo Fisher's Amira [31] is also a popular choice for image visualisation and annotation. It is particularly appreciated for its easy-to-use labeling tools to improve manual annotation and tracing, such as label interpolation, and image-guided picking. Amira's label interpolation was the main inspiration behind `napari-label-interpolator`, and image-guided picking is in the future plans for `blik` development. Being an image-focused tool, Amira is more limited when it comes to particle coordinate generation although it has some model-based filament picking that can be repurposed for particle generation with the help of user scripts. However, Amira's closed-source proprietary nature is a big downside for open science practices, making it hard or impossible to extend, contribute to, and freely share within the scientific community.

`EMAN2` [10] is a full software suite with an entire tomography pipeline from raw data to reconstruction. I/O compatibility between `EMAN2` and `blik` is partially implemented, allowing to read particles and tomograms. `EMAN2` has some tools for visualisation and picking, and is especially powerful for automated picking and segmentation thanks to ML tools. Its 3D visualisation is similar to `IMOD` in features.

Tomviz [32] is an open-source application focusing on tomogram reconstruction and visualisation, providing also a few segmentation and analysis tools.

When it comes specifically to particle picking and visualisation, a powerful tool recently developed is `ArtiaX` [33], a `ChimeraX` [24] extension for cryo-ET. Thanks to `ChimeraX`'s beautiful ray-tracing renderer, `ArtiaX` is ideal to make figures for publications. Particle visualisation is also very convenient and allows even for visualising subtomogram averaging results (map isosurfaces) distributed on the tomograms thanks to a performant implementation with instanced rendering. `ChimeraX` also provides a Python API to control its visualisations, but does not offer the same level of two-way and direct access to the visualised data as `napari` with its IPython console. `ChimeraX`'s features and ecosystem are also more focused on protein structure visualisation and analysis, whereas `napari` is first and foremost an imaging tool.

Another tool that `blik` already integrates with is `CrYOLO` [4], which provides powerful ML picking and segmentation routines. `CrYOLO` itself has recently adopted `napari` as its visualisation front-end.

A similar tool to the surface resampling widget in `blik` exists in `Membranorama` [34,35], which allows for visualising surfaces in a tomogram with the surrounding volume projected perpendicularly onto the surface, as well as extracting individual surface patches which can be "planarised" for easier inspection. This tool is very useful for interactive visualisation and exploration of the projected surface `in situ`. `blik`, on the other hand, does not allow `in situ` projection, but instead focuses on generating a square-grid resampling which—differently from `Membranorama`—can be exported as an ordinary volume or reused immediately for further processing with `blik` or other tools.

Given the popularity and fast growth of Cryo-ET, the field offers many other tools and software suites with features and goals compatible with `blik`. Some of of them offer excellent opportunities of integration with the `napari` and `blik` ecosystem, such as `TomoSegMemTV` [36] and `MemBrain` [37,38], which offer pixel-based and automated membrane segmentation (as opposed to the manual, surface-based annotation provided by `blik`) or `pycurv` [39] and `surface-morphometrics` [40], which can compute meshes, measurements, and statistics about pre-annotated membranes. With a collaborative and modular approach to software development, we strive for `blik` and `teamtomo` to become a starting point to enable such integrations in the future.

## Conclusions

The work presented in this paper aims to reduce the friction of working with cryo-ET data and to enable developers in the field to share, reuse, and contribute as a community. The development of `blik` and its components is a stepping stone towards these goals.

Working within `napari` allows us to delegate (and share with other fields) many non-cryo-ET-specific components, while retaining interactivity and extensibility; `napari` is in rapid development, and direct contributions to the community-developed project are always welcome. Even where direct contributions are unfeasible, developers can take advantage of the plugin ecosystem (such as `blik` does) or simple scripting.

Now that `blik`'s core features are established, we aim to reach out to other developers and cryo-ET software users and encourage reusing, adopting, or contributing to the work here presented.

Future planned features for `blik` include:

- Exploit `napari`'s nD visualisation, for example, to easily view the progression of a particle refinement.

- Conclude the work on `napari` multicanvas, allowing multiple views on the data (e.g., picking in orthoviews).

- Implement instanced rendering in vispy to allow rendering full particle maps in the tomogram at high performance (like `ArtiaX`).

- Offer more geometric models for picking (e.g., spheres, 3D lattices).

## Materials and methods

Not all the code contributions from this work live in the same place. Tools and implementations were split into standalone libraries or contributed to core `napari` when possible, in the interest of sharing and avoiding code duplication. Contributions from work in this paper are summarised below, and readers are encouraged to take advantage of all these open-source components for their own work.

Refer to the individual repositories for the most up-to-date and in-depth documentation.

### Cryotypes and cryohub

The data structures and IO functions used by `blik` are extracted into 2 usage-agnostic libraries: no assumptions from `blik` are carried over, which makes these libraries suitable for adoption by any Python software working with cryo-EM and cryo-ET data. Both libraries live in the github community project `teamtomo`.

- `cryotypes`: Defines simple and extensible data structures for cryo-EM data types and metadata, and provides simple validation and checking functions to ensure a given object conforms to the specification.

- `cryohub`: Provides reading and writing functions for popular image formats and particle data, with both fine-grained controls and a higher level "magic" interface (`cryohub.open(<anything>)`). Data is read to and from `cryotypes` data structures, easily allowing for conversion between formats and integration in any third-party Python tool.

  `cryohub` provides granular I/O functions such as `read_star` and `read_mrc`, which will all return objects following the cryotypes specification.

```python
from cryohub.reading import read_star

poseset = read_star('/path/to/file.star')
```

A higher-level function called `read` adds some magic to the IO procedure, guessing file formats and returning a list of `cryotypes`.

```python
from cryohub import read
data = read(
    '/path/to/file.star',
    '/path/to/directory/',
    lazy = False,
    name_regex = r'tomo_\d +'
)
```

See the help for each function for more info.

Similarly to the `read_*` functions, cryohub provides a series of `write_*` functions and a magic higher-level `write` function.

```python
from cryohub import write

write([poseset1, poseset2], 'particles.tbl')
```

## Morphosamplers

Surface and filament picking and particle generation code are not specific to cryo-ET. They were developed as part of a field-agnostic library called `morphosamplers`, which collects

several tools for sampling image data with morphological objects. As part of this work, models for spline filaments and spline-based surfaces were developed, with their relative tools for particle generation and image resampling.

## Napari plugins

Some of the `napari`-specific functionalities developed for `blik` were also not cryo-EM-specific and could instead be useful for many other applications in the `napari` ecosystem. These were extracted into their own `napari` plugins, which can be installed separately.

- `napari-label-interpolator`: A simple utility to interpolate n-dimensional labels along a specified dimension. Its main use in the context of cryo-ET is to reduce the manual annotation necessary to fully segment a volume. However, such functionality can also be used for example to track objects such as cells over a 2D (or 3D) time series.

- `napari-properties-plotter`: Several napari layers such as Points and Labels can hold features data for each of their items. This plugin allows to display any of such feature combinations in an interactive plot widget; users can then select a subset of the data items based on a selected section of the plot.

## Napari and vispy

Where possible, `napari`-specific code was contributed upstream to the `napari` core repository (or `vispy` for rendering-related code). Much of this work was distributed and collaborative in nature; here are listed some highlights that were crucial for the proper development of `blik` and to which significant contributions were made as part of this work.

- Improvements to quality, performance, and interactivity of 3D rendering for volumes and labels, including work such as proper depth buffer and blending usage, arbitrary plane slicing and clipping, additional 2D and 3D interpolation modes.

- Visualisation of points as spheres for more intuitive 3D visualisation.

- Improvements to surface mesh visualisation (shading).

- Projection of n-dimensional bounding box instead of simple point-slicing.

## Blik

Any remaining functionality specific to `napari` and cryo-EM was implemented directly in `blik` by often wrapping the aforementioned tools.

Manual particle picking makes use of the simple point picker in `napari`, while automatically adding orientations and metadata needed for writing to file.

Surface and filament picking are mainly wrappers around `morphosamplers`, but add a GUI for setting parameters and use `napari` layers for picking. The manual picks can then be used to generate visualisations such as filaments and meshes, for particle picking for subtomogram averaging, and for volume resampling.

A few image filtering and processing tools are also provided with `blik` for ease of visualisation, such as bandpass filtering and a power spectrum generator.

## Supporting information

**S1 Tutorial blik tutorial. A detailed tutorial for blik, describing the basics of the user interface and how to use essential features for practical applications.**
(PDF)

## Acknowledgments

We are particularly grateful to Alister Burt for stimulating discussions, suggestions, and initial guidance.

## Author Contributions

**Conceptualization:** Lorenzo Gaifas, Irina Gutsche.

**Funding acquisition:** Joanna Timmins, Irina Gutsche.

**Investigation:** Lorenzo Gaifas.

**Methodology:** Lorenzo Gaifas, Irina Gutsche.

**Resources:** Irina Gutsche.

**Software:** Lorenzo Gaifas.

**Supervision:** Joanna Timmins, Irina Gutsche.

**Validation:** Lorenzo Gaifas, Moritz A. Kirchner.

**Visualization:** Lorenzo Gaifas, Moritz A. Kirchner, Irina Gutsche.

**Writing – original draft:** Lorenzo Gaifas, Irina Gutsche.

**Writing – review & editing:** Lorenzo Gaifas, Moritz A. Kirchner, Joanna Timmins, Irina Gutsche.

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
