## [Editor Report · Decision Letter 0]

22 Nov 2023

Dear Irina, 

Thank you for submitting your manuscript entitled "blik: an extensible napari plugin for cryo-ET data visualisation, annotation and analysis" for consideration as a Methods and Resources Article by PLOS Biology. Please accept my sincere apologies for the delay in getting back to you as we consulted with an academic editor about your submission. 

Your manuscript has now been evaluated by the PLOS Biology editorial staff, as well as by an academic editor with relevant expertise, and I am writing to let you know that we would like to send your submission out for external peer review.

Once your full submission is complete, your paper will undergo a series of checks in preparation for peer review. After your manuscript has passed the checks it will be sent out for review. To provide the metadata for your submission, please Login to Editorial Manager (https://www.editorialmanager.com/pbiology) within two working days, i.e. by Nov 24 2023 11:59PM.

Kind regards,

Richard

Richard Hodge, PhD

rhodge@plos.org

PLOS

---

## [Decision Letter · Decision Letter 1]

22 Dec 2023

Dear Dr Gutsche,

Thank you for your patience while your manuscript "blik: an extensible napari plugin for cryo-ET data visualisation, annotation and analysis" was peer-reviewed at PLOS Biology as a Methods and Resources Article. Please accept my sincere apologies for the delays that you have experienced during the peer review process. Your manuscript has now been evaluated by the PLOS Biology editors, an Academic Editor with relevant expertise, and by two independent reviewers. 

In light of the reviews, which you will find at the end of this email, we would like to invite you to revise the work to thoroughly address the reviewers' reports.

As you will see, the reviewers are positive about the blik data visualization tool and think that the software will be useful and impactful for the cryo-ET field. However, Reviewer #1 raises concerns with the usability of the tool and thinks that a step-by-step tutorial should be included to make the software more user-friendly. Further to this point, the editorial team also noted that the manuscript text is currently aimed at a computational audience. We think it would be beneficial to reframe and edit the manuscript text with a general and broad biologist readership in mind. In addition, the reviewers note that additional capabilities should be included to increase its impact, such as the plotting of geometric shapes in the particle section.

Given the extent of revision needed, we cannot make a decision about publication until we have seen the revised manuscript and your response to the reviewers' comments. Your revised manuscript is likely to be sent for further evaluation by all or a subset of the reviewers.

**IMPORTANT - SUBMITTING YOUR REVISION**

*Re-submission Checklist*

*Published Peer Review*

*PLOS Data Policy*

*Blot and Gel Data Policy*

Sincerely,

Richard

Richard Hodge, PhD

rhodge@plos.org

REVIEWS:

Reviewer #1: In this manuscript, Gaifas et al describe a new Napari-based tool for visualisation, annotation and analysis of cryo-ET data, which they name blik.

Their software aims to be compatible with commonly used cryo-ET processing packages and provide a comprehensive suite of useful tools. As such, blik should significantly lower the energy required for projects where switching between different software is necessary, and would be a very welcome addition to the fast-growing cryo-ET field.

The paper is clearly written, however, some clarifications are needed. Some additional capabilities and tutorials should also be included before publication. Details of required revisions below.

Page 2, line 18-19. emclarity should be cited, as well as pytom/the av3/tom toolbox. While some of these are becoming obsolete, many of the tools currently used are still based on file formats from these software. Also the authors cite imod but also peet should be included.

Page 2, line 53 (and further references later). The authors refer to relion, but they need to clarify which relion. The current warp/M pipeline uses relion 3, while a self-sufficient modern subtomogram pipeline is offered in relion 4 and 5, however these are not compatible in terms of file format, and blik needs to be able to read and write star file compatible with relion 3 (to maintain compatibility with Warp/M) as well as 4/5.

Page 4, line 108. Fig 1E shows a slice through a power spectrum, not a gaussian filtered image as described.

Figure 2 and 'Particles' section. The depiction is beautiful and clear. However, there are tools out there such as 'place objects in space' from the Briggs lab which are widely used (and are based on the tom/av3 motivelist format), which currently offer to plot not just spheres and arrows, but also other geometrical shapes such as square, rectangles, triangles, hexagons etc. This is incredibly useful and should be added to blik.

Moreover, it is possible to plot any map (such as the subtomogram average). I appreciate this is described as part of the future developments in the discussion, however the authors only cite the recently released ArtiaX as their inspirations, while the tool has been freely available from the Briggs lab since 2018. 

Filaments section. This should include citations to the works from Briggs which also include filament pickers working in a very similar manner. (and which are also provided for spherical objects). Both here and in the surface section, more explicit reference to dynamo should be made. I appreciate a general statement of 'inspiration' is present in the discussion, but I think the authors should specifically mention which tools their ideas are based on.

Resampling: again a very nice tool, which is great to have within the same suite as all the others, but the section lacks reference to previous work from Dimitry Tegunov (membranorama).

Lastly, but most importantly, I have installed blik and had a quick play with it, and I find that it is not intuitive at all, especially for users who are not familiar with Napari (which is most people). For example, I was not able to work out how to display isosurfaces (which I should hope is possible). A step-by-step tutorial is required. 

Reviewer #2: Summary:

This manuscript describes the features of blik, an open-source Napari plugin for the visualization and annotation of cryo-electron tomography (cryo-ET) data. The authors demonstrate how blik builds on available functionality in Napari for cryo-ET data visualization by adding new features, such as 3D volume filters (e.g., gaussian filter, 3D power spectrum), particle data (e.g., location, orientation), image segmentation (e.g., volumetric voxel labeling), filament tracing, and surface generation. The authors demonstrate how output/input files from disparate cryo-ET software (e.g., dynamo, relion, imod) can be easily read and opened in blik. The authors showcase filament- and surface-based particle picking and resampling, akin to other methods, but distinguishably advantageous in blik due to its Python-based scripting.

Significance:

This manuscript tackles a crucial requirement in the cryo-ET field: the development of software that seamlessly integrates the visualization of output data from diverse sources into a unified platform. Due to the 3D nature of tomographic data, it is often difficult to visualize complex spatial relationships between different subcellular structures simultaneously. Napari visualization software laid the groundwork for handling cryo-ET data, and blik expands upon this foundation by integrating additional data analysis outputs commonly associated with such studies (e.g., segmentations, particle information, etc.). While there are still additional functionalities that the tomography field will likely want to incorporate, this software represents precisely the kind of tool the field desperately needs. I believe it marks an essential first step in the right direction.

Suggestions for improvement

Overall, this manuscript does a good job of describing the various functionality of blik with specific examples of the implementation of its different features visualized in the corresponding figures. Below are a few suggestions to improve the overall clarity and flow of the manuscript:

1. All the figure legends could benefit from more clarity. For example, figure 5 cannot stand alone with the current legend without referencing the relevant main text portion. The manuscript would also benefit from additional references to specific panels of the figures in the main text. Additionally, some of the figure panel references seem to be in the wrong place; for example, a reference to Figure 1E in line 108 may be better suited for the later section in line 119. 

2. The properties plotter plugin is super useful and exciting; however, the example given in Figure 2 may not be the best representation of the powerful utility of this tool. Index versus rlnLogLikeliContribution value seems like a poor representation that does not have much intuitive value. Perhaps it would be better to show a histogram of the values that give clear selection or scoring criteria, much like what is implemented in other software such as Pytom or Artiax. The authors mention in a later section, starting in line 267, "Particles may hold extra metadata such as classification results or quantitative values such as confidence score." It would be better to show a parameter like this in Figure 2. 

3. Related to the previous point, Lines 267-273 seem better suited in the earlier Particles section.

4. The discussion is relatively thorough in comparing Blik to other currently available cryo-ET software for simultaneous visualization of tomograms, segmentations, particle picks, etc. However, I believe it is worth mentioning other available software for generating and visualizing filament and membrane segmentations (i.e., tomomemsegtv, membrain-seg, dragonfly), density resampling (i.e., membranorama), and surface mesh reconstructions (i.e., pycurv, surface morphometrics).

---

## [Decision Letter · Decision Letter 2]

28 Mar 2024

Dear Dr Gutsche,

Thank you for your patience while we considered your revised manuscript "blik: an extensible napari plugin for cryo-ET data visualisation, annotation and analysis" for publication as a Methods and Resources Article at PLOS Biology. This revised version of your manuscript has been evaluated by the PLOS Biology editors, the Academic Editor and the original reviewers.

Based on the reviews, I am pleased to say that we are likely to accept this manuscript for publication, provided you satisfactorily address the remaining point raised by Reviewer #2. Please also make sure to address the following editorial and policy-related requests that I have provided below (A-B):

(A) We would like to suggest the following modification to the title:

“Blik is an extensible 3D visualization tool for the annotation and analysis of cryo-electron tomography data”

(B) Thank you for already depositing the source code for bilk in Github (https://github.com/brisvag/blik). However,

please note that we cannot accept sole deposition of code in GitHub, as this could be changed after publication. Instead, we ask that you please archive this version of your publicly available GitHub code to Zenodo. Once you do this, it will generate a DOI number, which you will need to provide in the Data Accessibility Statement (you are welcome to also provide the GitHub access information). See the process for doing this here: https://docs.github.com/en/repositories/archiving-a-github-repository/referencing-and-citing-content

We expect to receive your revised manuscript within two weeks. 

*Published Peer Review History*

*Press*

Kind regards,

Richard

Richard Hodge, PhD

rhodge@plos.org

Reviewer remarks:

Reviewer #1: I am satisfied with the improvements to the manuscript, and particularly welcome inclusion of a tutorial. I recommend publication.

Reviewer #2 (Danielle Grotjahn, signs review): I am satisfied with the revisions made, including the incorporation of suggested improvements. As a result, I recommend accepting this article for publication. One very minor suggestion: The authors should consider adding citations for the surface reconstruction programs pycurv and surface-morphometrics (line 375).

---

## [Editor Report · Decision Letter 3]

2 Apr 2024

Dear Dr Gutsche,

On behalf of my colleagues and the Academic Editor, David Bhella, I am pleased to say that we can accept your manuscript for publication, provided you address any remaining formatting and reporting issues. These will be detailed in an email you should receive within 2-3 business days from our colleagues in the journal operations team; no action is required from you until then. Please note that we will not be able to formally accept your manuscript and schedule it for publication until you have completed any requested changes.

PRESS

Best wishes, 

Richard

Richard Hodge, PhD

rhodge@plos.org

PLOS
